# Development of Welfare Protocols at Slaughter in Farmed Fish

**DOI:** 10.3390/ani14182730

**Published:** 2024-09-21

**Authors:** Raffaelina Mercogliano, Alessandro Avolio, Floriana Castiello, Maria Carmela Ferrante

**Affiliations:** Department of Veterinary Medicine and Animal Production, University of Naples Federico II, Via F. Delpino 1, 80137 Napoli, Italy; alessandro.avolio@unina.it (A.A.); florianacastiello@gmail.com (F.C.); mariacarmela.ferrante@unina.it (M.C.F.)

**Keywords:** farmed fish, slaughter, post mortem process, fish quality, welfare protocol

## Abstract

**Simple Summary:**

Slaughter and pre-slaughter practices can impact the flesh quality of farmed fish in a significant manner. Fish are sentient animals, able to experience emotions and psychological stress. They can perceive pain when killing methods are applied. European legislation does not adequately protect the welfare of farmed fish, unlike other animals. This study discussed how the implementation of slaughter and killing protocols, which include welfare indicators, can effectively enhance the well-being of fish and the quality of fish meat.

**Abstract:**

The study investigated fish welfare at slaughter. Killing animals may induce suffering to the animals even under the best available technical conditions. Moreover, fish have different physiological characteristics and are slaughtered differently from terrestrial animals. The use of commercially available methods exposes farmed fish to pain and suffering during slaughter, which could lead to acute stress and post mortem changes in fish quality. The study aimed to discuss (i) the current knowledge and knowledge gaps on fish welfare related to stunning and killing methods; (ii) the variables that affect the post mortem changes in fish meat, and (iii) the indicators of welfare during slaughter. Application of welfare protocols at slaughter improves fish welfare. Specific protocols for fish are not provided in EC Regulation 1099/2009 on animal protection at killing. Detailed guidelines in the fish welfare assessment may allow the development of specific fish legislation. Developing humane technologies might have important effects on fish quality, consumer perception and aquaculture economics.

## 1. Introduction

Consumers increasingly demand information about the origins of food products. They judge the quality of animal products considering the ethic production strategies, respect of animal welfare, and potential consequences on food safety [1]. 

Even under the best technical conditions, killing animals can still inflict pain, fear, and suffering. Stunning should result in an immediate unconsciousness and insensibility state that continues until death, and if it is not possible for an instantaneous induction, it should be carried out without causing avoidable suffering. 

In a good state of welfare, an animal is healthy, comfortable, well nourished, and shows the innate behavior not experiencing pain and fear [2]. This definition applies to all species of animals including fish farmed for food [3]. Moreover, the concept of fish welfare is rather complex compared to terrestrial animals. 

As sentient animals, fish can experience emotions, fear, psychological stress and pain [4,5,6]. Evidence for the neural components of sentience in fish has been demonstrated by research on their sensory systems, brain structure and functionality [3]. Emotional memory in fish is related to the function of the medial pallium region [7]. This is homologous to the mammalian amygdala and is part of the limbic system. 

Fish have well-developed pain pathways that, in some species, involve higher brain centers such as telencephalon. In response to noxious stimuli, fish cognitively perceive pain which has important fitness functions, and experience positive and negative emotions that provide insights into their welfare [8,9,10]. 

Nevertheless, European legislation does not adequately protect the welfare of farmed fish [11,12]. European Council Regulation (EC) 1099/2009 on the protection of animals at slaughter fixed specific requirements for the terrestrial species farmed for food. Unlike for other farmed species, moreover, the regulation did not define stunning methods for fish, only recommending fish protection standards at killing (Annex 1). Thus, despite physiological differences between fish and terrestrial animals, the provisions applicable to fish welfare at slaughter are limited to the basic principles (Art. 3): (a) during slaughter and related operations, any avoidable pain and suffering shall be spared; (b) farmed fish should be killed rapidly with minimal fear and pain [13]. In addition, although the regulation permits the European Member States to adopt national rules for protecting fish at the time of killing (Art. 27), only a few States have already enacted national laws that only pertain to a few species (like salmon) [14,15]. In order to address the law gaps, the Farm to Fork strategy of the European Green Deal will revise the (EC) Regulation 1099/2009 considering welfare criteria and the publication of specific guidelines for farmed fish [16].

When killing methods are applied in farmed fish, stunning and slaughter may play a role as acute stressors. During slaughter, stress-related pathways can interact with post mortem metabolic processes, leading to changes in fish meat quality traits such as accelerated post mortem metabolism, low ultimate pH, impaired water-holding capacity, changes in texture and meat color, and more rapid meat degradation [17].

This study focused on the context of farmed fish welfare at slaughter, highlighting (i) the current knowledge and knowledge gaps on the available stunning and killing methods; (ii) the stressors influencing the post mortem changes in fish meat, and (iii) the welfare indicators at slaughter. 

## 2. The Context of Welfare at Slaughter in Aquaculture

Slaughter is the process used for the killing of animals intended for human consumption [13]. Safety methods for stunning and killing fish have been developed to achieve quality control of products. The available commercial methods include electrical stunning followed by decapitation, percussive stunning using a captive bolt followed by gill-cutting, the use of carbon dioxide (CO_2_) as a stunning method, and the application of water slurry as a killing method.

### 2.1. Welfare at Slaughter in Farmed Fish

Different fish species require different technical approaches to stunning and killing procedures. Staff training and technical capabilities are essential for handling and slaughtering components and may require changes in operational protocols. Animals should only be killed after stunning, and the loss of consciousness and sensibility should be maintained until death [13]. However, effectively stunning has evident benefits for welfare and product quality, including painless killing, better flesh quality, delayed rigor mortis, and longer shelf life [18]. It is crucial to determine if the methods can be either stressful or painful for farmed fish [19]. Among the stunning methods, in the application of electrical stunning (in water or after fish are removed from the water), a sufficient current should be passed through the fish’s brain using an adequate amperage across the electrodes after fish removal from the water. The waveform, the stunner orientation, and fish density (kg/L) influence the effectiveness of an electrical stun. Stunning induces a loss of consciousness in fish, but its effectiveness can significantly vary depending on the parameters set (voltage (V), ampere (A), hertz (Hz), and time of application.

To establish the exposure duration to the electricity, the main requirements that must be established are the time interval between fish leaving the stunner and the killing method [19].

Percussion is a stunning method that can be applied manually or automatically through a blow to the fish’s head. When the automated percussive method is applied, variations in the size within the fish population can cause a mis-stun in some individuals, and the fish quality can be affected. In this case, a further killing method must be applied. Controlling the process requires the presence of well-trained staff at the slaughter facility. The air pressure method used to drive the bolt should be checked to ensure that it is sufficiently high and that the blow is delivered correctly to the head of each fish. Percussion followed by a correct gill-cutting causes a rapid status of unconsciousness and insensibility before slaughtering [3,4,5,6,7,8,9,10,11,12,13,14,15,16,17,18,19]. If compared to CO_2_ stunning, the implementation at commercial slaughterhouses of percussive stunning seems to improve the welfare of species such as Atlantic salmon. Spiking/coring underwater and underwater shooting with *Lupara* are the preferred methods for tuna [3,4,5,6,7,8,9,10,11,12,13,14,15,16,17,18,19,20].

As a killing method, ice water slurry (a mixture of ice and water from 1:2 to 3:1) is used for rainbow trout, European sea bass and gilthead seabream. The transfer into ice water may lead to asphyxia due to the increased fish density without any aeration or oxygen addition to the slurry. The method poses a severe welfare risk and is ineffective in species surviving long periods of anoxia. For example, the rapid drop in temperature has a higher impact than the oxygen absence in Crucian carp, which is more tolerant to low oxygen levels [19]. Without prior stunning, asphyxiation on ice is considered the worst method in relation to welfare and flesh quality [3] (Table 1).

In the current context, some issues are critical for fish welfare at slaughter. Stunning should induce instantaneous and long-lasting unconsciousness allowing personnel sufficient time to kill fish. Moreover, certain practices applied to farmed fish could lead to injury or damage of tissues [10]. The stunning application might be ineffective when immobility occurs before consciousness is lost. In this case, there is a risk that exsanguination and evisceration are performed on a conscious animal. If the period of unconsciousness is too short, fish may recover consciousness before being killed. 

New stunning methods are proposed to improve the welfare of farmed fish. As an innovative stunning/killing method, immersion in cold saline (−6 °C, 5% NaCl) is considered an alternative to percussion in rainbow trout [21]. Also, in the same species, electronarcosis using a current intensity of 400 mA is proposed as an alternative to fish immersion in ice water, reducing the stress response and improving flesh quality [22]. 

Nevertheless, methods to make fish unconscious are still being used in a few slaughterhouses. Asphyxiation in the air or an ice slurry, gill cutting, CO_2_ addition, exsanguination in ice water, salt baths, or ammonia application are stressful and may induce post mortem changes in fish meat. Finally, in aquaculture, the use of more humane methods (percussion, electric stunning) are restricted only to certain fish species [10]. 

### 2.2. Post Mortem Changes in Fish Meat under Stress

Blood circulation and oxygen supply cease after the fish dies, and glycogen can no longer be oxidized. To maintain energy expenditure, when ATP reserves are depleted, the breakdown of glycogen occurs through muscle anaerobic glycolysis. The lactic acid accumulation generates H+ ions and lowers pH values until all glycogen is consumed. These reserves of energy are influenced by stress intensity and duration while fish are alive [23]. 

In aquaculture plants, fish deal with stressful situations (stressors) that can compromise their welfare. A cascade of reactions is involved in the stress response, enabling the fish organism to cope the stressors.

At slaughter, the most probable mechanisms and responses arising under stress in fish, mammals, and poultry are analogous. Adrenergic (rapid) stress response mediated by the sympathetic nervous system and the release of catecholamine (adrenaline and noradrenaline) can be caused by perceived stressors like the presence of a predator, physical stressors (e.g., handling, transportation) or chemical stressors (e.g., low oxygen concentration, increased acidity).

Due to the function of the hypothalamic-pituitary-adrenal axis, stressors may also induce a long-term neuroendocrine response. The release of adrenocorticotropic hormone leads to the production of cortisol from the adrenal cortex [17,18,19,20,21,22,23,24].

Usually, the stress response in fish induces an increase in the muscle activity (swimming, wrestling, escaping behavior), leading to a reduction in the energy reserves of muscle. Although a strong post mortem drop in pH occurs in mammalian (ultimate pH 5.4–5.8) and poultry meat (ultimate pH 5.6–6.1), since meat is poor in glycogen, the post mortem pH decrease is significantly slighter (ultimate pH > 6.0–6.2) in fish [17].

Repeated or prolonged stress conditions will lead to the clearing of lactic acid and the gradual exhaustion of the energy reserves. The absence of substrate for anaerobic glycolysis will result in high pH values for fish killed during the period of exhaustion. In flesh-stressed fish, pH does not change significantly until 24 h, while in anaesthetized fish, it gradually decreases, reaching ultimate values at 18 h post mortem. Due to accelerated glycogenolysis, high blood levels of cortisol and plasma glucose concentrations might be responsible for these changes [17]. 

Texture is an important property of fish muscle. If compared to mammalian meat, in fish meat, there are significant differences in the activity of proteolytic enzymes and the rate of muscle protein degradation. Due to the lower muscle glycogen level, after rigor mortis, proteolysis in fish quickly induces muscle relaxation after slaughter. Cathepsins are the main proteases responsible for the myosin degradation. Cathepsin B and L degrade a-actinin, troponin T and desmin [24]. Fish species may have different mechanisms of post mortem degradation. Desmin is degraded in sardine and turbot, and not in sea bass and brown trout. In addition, other endogenous proteases, such as trypsin-like, collagenase-like enzymes and metalloproteases, might be responsible for the proteolytic processes in other fish species [24]. 

Fish consumers prefer a firm meat texture that depends on water-holding capacity. When combined stress factors occur, fish meat shows significantly reduced water-holding capacity, elevated muscle collagenase-like activity immediately post-slaughter, and hardness after some days of ice storage. Water-holding capacity and fish texture are related to pH decline, proteolytic processes, and solubility of muscle protein. Significant post mortem changes (often accelerated occurrence of the processes) in fish muscle and some changes in the meat quality traits arise under acute short-lasting (minutes, hours) stress [25] (Table 2).

The rapid depletion of ATP and pH values increases the concentrations of metamyoglobina which may induce a faster post mortem discoloration (redness/yellowness) in fish meat [17].

## 3. Welfare Protocol Development at Farmed Fish Slaughter 

In the fishing industry, animal welfare is not only an ethical perspective. Developing animal welfare standards and implementing sustainable techniques represent opportunities to promote consumer trust and economics in aquaculture [26]. To implement welfare protocols, the first step is setting the stunning conditions to obtain an effective stun in a laboratory. In the second step, the impact on meat quality of the stunning/killing methods are evaluated. In this step, laboratory (stress and neurophysiological) measurements are not feasible during the procedure implementation. Thus, in a commercial setting, behavioral and physical measures will be carried out. The third step considers the application of systems of quality assurance to monitoring the slaughter processes in practice [19].

### 3.1. Welfare Indicators at Slaughter

The application of an integrated approach is proposed (a) to establish specifications at slaughter (i.e., by stunning them before killing) in a laboratory setting and (b) to assess their effective implementation in practice [27].

Based on this approach, welfare indicators (WIs) at slaughter are identified. They are measures providing information about the level of welfare fulfilment of animal that may be related to their welfare state. WIs describe the welfare degree considering fish species and specific life stages under the farming conditions [19]. Welfare protocols include Laboratory-based Welfare Indicators (LABWIs), Operational welfare indicators (OWIs), and input- and outcome-based WIs [28]. 

LABWIs evaluate insensibility and onset and duration of unconsciousness, assessing the effects of a stunning method on the fish brain function [26]. LABWIs which unequivocally assess the brain function for recognizing fish unconsciousness in the laboratory utilize electroencephalogram (EEG), associated with the observation of sensory evocated responses (SERs) and visual evoked responses (VERs), and electrocardiogram (ECG) [29]. EEG determines whether the stunned fish can be roused by monitoring the brain’s electrical activity. Characteristic EEG changes (e.g., a general epileptiform insult) induced by pain stimuli are indicative of unconsciousness. The absence of fish response indicates that unconsciousness and insensibility last until death. SERs, such as the brain response to a pain stimulus, and VERs, such as the response caused by the flashes effect of a light directed towards the eyes, should be registered. The insensibility state is indicated by the absence of SERs on EEG trace, while an average VERs absence indicates brain dysfunction. ECG describes the heart’s electrical activity, providing additional information for the welfare assessment [30].

Physiological LABWIs of stress post mortem are parameters such as the level of blood metabolite (e.g., glucose, lactate, hepatic glycogen, heat shock proteins, metabolically active enzymes), immune system function (e.g., the number of circulating lymphocytes), hormones and neurotransmitters determination, corticosteroid receptors abundance, and specific gene regulatory pathways. The cortisol level measure indicates an arousal increase and may be considered to assess stunning/killing methods [28]. Finally, a promising approach to the welfare assessment is the measure of cortisol, melatonin and Acetyl-N-formyl-5-methoxykynurenamine from the fish-secreted mucus. The evaluation of this substance in fish mucus represents the expression of a local (cutaneous) system of stress response (CSRS) analogous to that of mammals [30].

OWIs are useful on-farm measures. They may be carried out on the animal (e.g., behaviors, skin condition); on the water (e.g., oxygen, temperature), or to evaluate management procedures (e.g., feeding). OWIs should be valid, reliable, repeatable, comparable, suitable and practical indicators. 

Input-based WIs evaluate the fulfilment of one or more animal welfare needs (e.g., treatments the animals are subjected to). Outcome-based WIs are direct welfare indicators based on measures of animal behavior, injuries or stress physiology. They describe the response of an animal to effects on its welfare state and are practical and feasible measurements (e.g., physical measurements) that should be used associated with behavioral observations and visual inspection. Outcome-based WIs can evaluate if stunning and killing equipment in practice is responsive to welfare requirements [19,20,21,22,23,24,25,26,27,28]. To verify the correct stunning and unconsciousness state, the main outcome-based WIs in fish are the loss of opercular activity, visual evoked response, vestibule-ocular reflex and eye-rolling [2].

### 3.2. Application of Welfare Indicators at Slaughter

Welfare assessment of slaughter must be based on values and scientific data considering all steps of the process [2]. Ideally, the farmer should be able to interpret OWIs. Moreover, a three-level system has been proposed to apply them at slaughter [31,32] (Figure 1).

Level 1 utilizes simple and rapid passive OWIs associated with observations of the fish’s appearance, behavior and mortality rate. At this level, if input-based WIs are not functional, the farmer should identify the causes and implement actions of mitigation. If outcome-based WIs are not functional, the farmers should be required to seek consultant advice from a fish health professional. In the absence of information to identify the cause, OWIs of level 2 and LABWIs can be immediately utilized by a farmer. Level 2 provides the study of the environment, rearing practices and handling, and sampling fish to identify possible causes. Level 3 should be considered when the information is not comprehensive. This level will provide more complex OWIs and LABWIs requiring the presence of expert skills (health professionals and technical personnel) [31,32].

At slaughter, to ensure the fish’s welfare, it is important that handling methods are applicable to the biological characteristics of fish and environment is an appropriate context to fulfil its needs. Correlated operations and holding facilities must be considered for the welfare protocol development. The personnel engaged should be qualified in fish handling, able to understand the behavior models, and the underlying principles to carry out their tasks. Training should include occupational health and knowledge on the food safety impacts of the slaughter methods [2].

Farmers need to monitor and assess the fish quality. If fish must be transported before stunning, the holding facilities are constructed to hold a specific species/group. The size of facilities must facilitate the slaughter processes of a given number of fish in a certain timeframe respecting the welfare requirements. Tanks, pumps pipes, and other equipment for transferring fish must be used to avoid physical injuries, and the quality of water must be suitable to the species and stocking density of fish. All handling, stunning and killing equipment should be tested regularly to ensure adequate performance [2]. The stunning and killing methods should consider species-specific information. The application of a backup stunning system is needed to verify the absence of consciousness. Fish should be re-stunned as soon as possible in case of mis-stun or regaining of consciousness before death [2]. If killing is to be delayed, stunning should not be carried out to avoid the recovery of fish consciousness.

## 4. Conclusions

Consumer perception can influence market demand, promoting the adoption of welfare protocols. A ComRes Institute for Eurogroup for Animals survey involving 9000 EU citizens has shown that 89% of them consider slaughter an important welfare criterion for farmed fish. Also, consumers consider animal-friendly products more authentic and environmentally friendly. 

The purpose of welfare protocols at slaughter is to define how slaughter methods must be applied and how slaughter equipment can be checked. This may ensure that the required welfare parameters are applied in practice. Higher-quality meat is obtained when species-specific methods are used and the stunning or killing parameters are correctly applied. During the procedure implementation in a commercial setting, stunning and killing should be monitored using a quality assurance system in which all the correlated operations must be conducted to minimize injury and stress. 

The EU Commission has proposed possible actions to improve the welfare of farmed fish at slaughter. The proposal has included species-specific provisions for the killing of Atlantic salmon, common carp, rainbow trout, European sea bass and gilthead sea bream; the review of current requirements not applicable to farmed fish; the introduction of a pre-approval system for stunning and restraining equipment, and the use of more appropriate slaughter equipment. Finally, the EU Commission has required, as preconditions for approval of slaughterhouses, the manufacturers’ instructions on the use of restraining and stunning equipment and adequate staff training quality.

The main limit to the applicability of the EC Regulation 1099/2009 is that it does not provide specific protocols for fish. Detailed guidelines in the fish welfare assessment may allow for the development of specific recommendations and legislation. 

Considering that European consumers perceive farmed fish as more ethical than wild fish, and animals and slaughter practices have effects on fish flesh quality, welfare improvement may promote fish well-being, consumer trust, and aquaculture economic activities.

## Figures and Tables

**Figure 1 animals-14-02730-f001:**
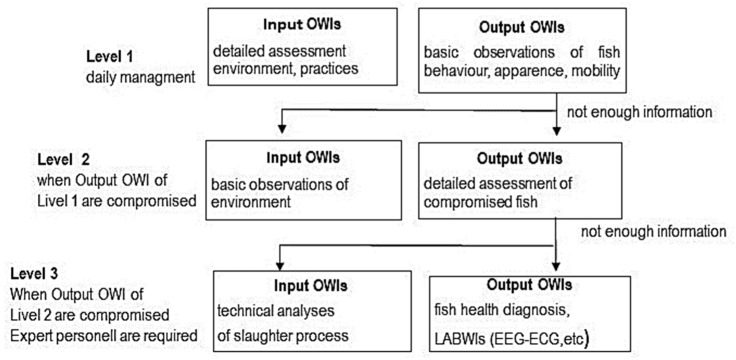
Three-level system for the OWIs application at fish slaughter (summarized from [32]).

**Table 1 animals-14-02730-t001:** Farmed fish welfare at slaughter. Effectiveness evaluation of current methods.

Method	Welfare Evaluation
Hemorrhagic decapitation, evisceration, exsanguination without a prior stunning method	rejected
Gas exposure immersion in N_2_/CO_2_ saturated water	not recommended for Atlantic salmon and sea bream
Isoeugenol anesthesia	not recommended for potential adverse effects
Full-automatic percussive stunning	rejected for overly high stunning failure rate
In-water pipeline electrical stunning followed by semi-automatic percussive method	acceptable ***
In-water pipeline electrical stunning followed by an efficient killing method	acceptable **
In head-to-body electrical stunning In-water rotating electrical stunning	acceptable only in freshwater *
Semi-automatic percussive stunning	acceptable only in freshwater if the design is adequate and guarantees an air exposure under 15”.

Legend: *** = first-** = second-** = 3° third best available method. Note: welfare evaluation based on [2].

**Table 2 animals-14-02730-t002:** Post mortem changes in fish meat under acute stress conditions (summarized from [17]).

Fish Muscle	Post Mortem Change
Pre mortem energy reserves depletion Lower initial pH-Ultimate pH Rigor mortis Accelerated/rapid autolysis Denaturation of muscle proteins Rapid ATP depletion Myofibril and connective tissue physical stress	accelerated post mortem metabolism final rapid pH increase rapid onset and early resolution reduced meat water-holding capacity liquid loss muscle physical damage activation of muscle proteases

## Data Availability

The raw data supporting the conclusions of this article will be made available by the authors upon request.

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
