# Peer review of "Development of Welfare Protocols at Slaughter in Farmed Fish"

_animals, 2024, doi:10.3390/ani14182730_

Round 1
Reviewer 1 Report
Comments and Suggestions for Authors
Writing a commentary paper on the development of welfare protocols at slaughter in farmed fish is important. The paper highlights the ethical responsibility to ensure the humane treatment of farmed fish, acknowledging that fish are sentient animals capable of feeling pain and stress. Also, Identifies current gaps in knowledge and areas where further research is needed, helping to drive scientific inquiries into better welfare practices.
Enhanced welfare measures can improve product quality by lowering stress-related issues, potentially enhancing market value and consumer acceptance. Improved welfare at slaughter can result in more efficient processing and fewer losses owing to fish death or poor quality.
The study has a limited scope, it primarily analyzes existing material without suggesting new experimental research or innovative approaches. The commentary provides a broad summary without going into detail about specific parts of welfare procedures or the complexity of physiological differences across fish species. Limited comparative analysis of various stunning and killing methods may impede the selection of the most effective welfare practices.
The discussion is fully addressing the economic and practical challenges of implementing new welfare protocols across diverse aquaculture operations.
The paper does not include perspectives from all relevant stakeholders, such as fish farmers, industry experts, animal welfare organizations, and consumers.
It has not adequately addressed how consumer perceptions and market demands influence the adoption of welfare protocols.
My recommendations are:
Include detailed case studies of successful implementation of welfare protocols in various aquaculture settings to provide concrete examples and practical insights.
Suggest or review recent experimental research on innovative stunning and killing methods to highlight potential future directions.
Provide a comparative analysis of different stunning and killing methods, assessing their relative effectiveness and potential for widespread adoption.
ritically evaluate the limitations and benefits of current methods, identifying specific areas where improvements are needed.
Conduct interviews with various stakeholders, including fish farmers, industry experts, and animal welfare advocates, to incorporate diverse perspectives and practical insights.
Offer detailed guidelines for welfare protocols that can be adapted to different species and regions, promoting a more nuanced approach to fish welfare.
Highlight emerging technologies and innovative methods that can improve welfare standards, encouraging further research and development in this area.
Reviewer 2 Report
Comments and Suggestions for Authors
This paper discusses the impacts of stunning and killing on fish welfare, indicators available to develop humane slaughter protocols and biochemical changes which occur in the carcass postmortem.
The main comment is that it requires greater clarity and precision. Terms such as distress, nociception, stunning, slaughter and consciousness need to be defined clearly and used consistently. Definitions of stress and distress from international sources are provided below.
(National Research Council (US) Committee on Recognition and Alleviation of Distress in Laboratory Animals. Recognition and Alleviation of Distress in Laboratory Animals. Washington (DC): National Academies Press (US); 2008. 2, Stress and Distress: Definitions. Available from: https://www.ncbi.nlm.nih.gov/books/NBK4027/)
Stress is an inferred internal state. A general distillation of the literature suggests that stress denotes a real or perceived perturbation to an organism’s physiological homeostasis or psychological well-being.
Distress - Most definitions characterize distress as an aversive, negative state in which coping and adaptation processes fail to return an organism to physiological and/or psychological homeostasis.
(https://www.nhmrc.gov.au/about-us/publications/australian-code-care-and-use-animals-scientific-purposes)
Distress - an animal is in a negative mental state and has been unable to adapt to stressors so as to sustain a state of wellbeing. Distress may manifest as abnormal physiological or behavioural responses, a deterioration in physical and psychological health, or a failure to achieve successful biological function. Distress can be acute or chronic and may result in pathological conditions or death
Line 143-144 contains a comparable reference to distress i.e. If a stressful event is repeated or prolonged, the organism fails to regain homeostasis impairing welfare.
The definitions of distress above do not fit with the way the word is used elsewhere throughout the manuscript. For example on Line 14-15 ‘Killing animals may induce dis-tress and other suffering even under the best available technical conditions.’ Whist killing an animal obviously prevents it from sustaining a state of wellbeing, killing is not a logical context to apply it. Suggest writing ‘Killing animals may induce suffering even under the best available technical conditions.’ For the same reason, I suggest removing the word distress elsewhere in the paper when associated with stunning or killing e.g. line 34, line 199
Line 45-47 ‘The consciousness presence in fish is related to the function of the medial pallium region which is homologous to the mammalian amygdala and is part of the limbic system responsible for emotional memory [7].’ The abstract of reference [7] states the following ‘In contrast, the medial pallium, considered homologous to the amygdala, is involved in emotional conditioning in teleost fish.’ Emotional conditioning and consciousness are not the same. Suggest writing ‘Emotional memory in fish is related to the function of the medial pallium region [7]. This is homologous to the mammalian amygdala and is part of the limbic system.’
Line 48-49 ‘Fish have a well-developed nociception that in some species involves higher brain centres such as telencephalon.’ The IASP (https://www.iasp-pain.org/resources/terminology/) defines nociception as ‘The neural process of encoding noxious stimuli. Note: Consequences of encoding may be autonomic (e. g. elevated blood pressure) or behavioral (motor withdrawal reflex or more complex nocifensive behavior). Pain sensation is not necessarily implied’. Pain is defined as ‘An unpleasant sensory and emotional experience associated with, or resembling that associated with, actual or potential tissue damage.’ Hence activity induced in the nociceptor and nociceptive pathways by a noxious stimulus is not pain. Linking nociception with higher brain centres such as telencephalon implies the experience of pain. Hence it may be more accurate to refer to pain pathways in line 48-49 ‘Fish have well-developed pain pathways that in some species involves higher brain centres such as telencephalon.’
Line 81 refers to carbon dioxide stunning as a commercial method of slaughter. Stunning is rendering unconscious. Slaughter is killing. Stunning and slaughter are not the same.
The paragraph from line 89-100 is confusing. Electrical stunning to induce insensibility before killing should be clearly distinguished from killing by electrocution. Electrical stunning of fish is clearly explained here - https://www.hsa.org.uk/humane-harvesting-of-fish-electrical-stunning/electrical-stunning-2. Hence line 90-91 should not refer to electrical stunning as a ‘stunning and killing method’. Line 100 should not refer to a ‘further’ killing method to follow electrical stunning. Heart fibrillation (line 94) may occur during electrical stunning but referring to it is distracting because stunning involves passing sufficient current through the brain to induce immediate unconsciousness and insensibility to pain. The term dewatering is confusing because it usually means removing water from structures or locations such as fish tanks, dams and construction sites rather than the process of removing fish from water.
Line 156-158 ‘Unlike mammalian and poultry meat, which show a strong post-mortem drop in pH, fish meat is poor in glycogen and the post-mortem pH decrease is significantly slighter (ultimate pH lower than 6.0-6.2).’ This compares a subjective statement (strong post-mortem drop in pH) with a semi-objective statement (ultimate pH lower than 6.0-6.2). However lower than 6.0-6.2 could mean anything down to 0. It would be better to specify exact pH ranges for each. E.g. ‘A strong post-mortem drop in pH occurs in mammalian meat (final pH range XX-YY) and poultry (final pH range XX-YY). Since fish meat is low in glycogen, the post-mortem drop in pH is less pronounced (final pH range is XX-YY).’
Comments on the Quality of English LanguageComments/suggestions about English are mainly about stylistic preferences. There are some typographical errors and confusing sections.
Line 11 ‘We talked about how’ suggest writing ‘This manuscript discusses about how’
Line 42-43 ‘able to perceive emotions experiencing fear, psychological stress and pain’. Suggest writing ‘able to experience fear, psychological stress and pain’
Line 57 ‘Thus, despite fish physiological differences from terrestrial animals’ suggest writing ‘Thus, despite physiological differences between fish and terrestrial animals’
Line 71 ‘and a faster loss of meat freshness’ suggest writing ‘and more rapid meat degradation’
Line 104 ‘A well-trained staff’ write ‘Well-trained staff’
Line 171-172 ‘Cathepsins are the main ones responsible for the degradation of the myosin.’ Write ‘Cathepsins are the main proteases responsible for the degradation of the myosin.’
Line 187 Caption ‘modified by’ should be ‘summarised from’
Line 219-220 ‘somatosensory evocated responses (SERs) and visual evoked responses (VERs) [26]’. This should be ‘sensory evoked-responses’ and the reference should be [27].
Lines 249-250 ‘Outcome-based indicators allow to assess’ suggest writing ‘Outcome-based indicators allow assessment of’
Line 261 It is not clear whether the reference in Table 2 (modified by Noble et al., 2018) is included in the bibliography. ‘Modified by’ should probably be ‘summarised from’.
Table 2 Livel 1, 2 and 3 should be Level 1, 2 and 3.
Line 278 ‘hold facilities’ should be ‘holding facilities’
Line 297 ‘The welfare protocols at slaughter must consider how slaughter parameters and methods must be determined’. This statement is confusing.
Line 299 ‘Higher-quality meat is obtained when specie-specific methods’ specie should be species.
Round 2
Reviewer 1 Report
Comments and Suggestions for Authors
The authors have provided the responses to the questions and the paper has been improved according to the comments